# Fat Consumption Attenuates Cortical Oxygenation during Mental Stress in Young Healthy Adults

**DOI:** 10.3390/nu15183969

**Published:** 2023-09-14

**Authors:** Rosalind Baynham, Samuel J. E. Lucas, Samuel R. C. Weaver, Jet J. C. S. Veldhuijzen van Zanten, Catarina Rendeiro

**Affiliations:** 1School of Sport, Exercise and Rehabilitation Sciences, University of Birmingham, Birmingham B15 2TT, UK; rxb585@student.bham.ac.uk (R.B.);; 2Centre for Human Brain Health, University of Birmingham, Birmingham B15 2TT, UK

**Keywords:** high-fat, mental stress, cortical oxygenation, near-infrared spectroscopy

## Abstract

Mental stress has been associated with cardiovascular events and stroke, and has also been linked with poorer brain function, likely due to its impact on cerebral vasculature. During periods of stress, individuals often increase their consumption of unhealthy foods, especially high-fat foods. Both high-fat intake and mental stress are known to impair endothelial function, yet few studies have investigated the effects of fat consumption on cerebrovascular outcomes during periods of mental stress. Therefore, this study examined whether a high-fat breakfast prior to a mental stress task would alter cortical oxygenation and carotid blood flow in young healthy adults. In a randomised, counterbalanced, cross-over, postprandial intervention study, 21 healthy males and females ingested a high-fat (56.5 g fat) or a low-fat (11.4 g fat) breakfast 1.5 h before an 8-min mental stress task. Common carotid artery (CCA) diameter and blood flow were assessed at pre-meal baseline, 1 h 15 min post-meal at rest, and 10, 30, and 90 min following stress. Pre-frontal cortex (PFC) tissue oxygenation (near-infrared spectroscopy, NIRS) and cardiovascular activity were assessed post-meal at rest and during stress. Mental stress increased heart rate, systolic and diastolic blood pressure, and PFC tissue oxygenation. Importantly, the high-fat breakfast reduced the stress-induced increase in PFC tissue oxygenation, despite no differences in cardiovascular responses between high- and low-fat meals. Fat and stress had no effect on resting CCA blood flow, whilst CCA diameter increased following consumption of both meals. This is the first study to show that fat consumption may impair PFC perfusion during episodes of stress in young healthy adults. Given the prevalence of consuming high-fat foods during stressful periods, these findings have important implications for future research to explore the relationship between food choices and cerebral haemodynamics during mental stress.

## 1. Introduction

Episodes of acute stress have been shown to trigger cardiovascular events [1,2,3], as well as stroke [4], potentially via stress-induced impairments in vascular function [5]. Cardiovascular health can also directly impact brain health, with well-established associations between lower cardiovascular diseases (CVD) risk and reduced rates of dementia and cognitive decline in older age [6,7]. Similarly, chronic stress affects brain function and cerebrovascular responsiveness, with stress altering the functional connectivity of the pre-frontal cortex (PFC) and impairing attention control in healthy adults [8], both known to be modulated by vascular health [9].

Acute laboratory mental stress has been evidenced to increase cerebral blood velocity [10], as well as increase oxyhaemoglobin and decrease deoxyhaemoglobin concentration in the PFC [11,12], both outcome measures indicative of elevated cerebral blood flow (CBF). Increased nitric oxide (NO) bioavailability and systemic increases in blood pressure and cardiac output have all been implicated as potential mechanisms for the increased cerebral perfusion induced during stress [12,13,14]. Importantly, CBF during mental stress is impaired in populations at risk of CVD, such as people with hypertension [15]. Therefore, it is likely that vascular dysfunction attenuates the cerebrovascular response to stress, which may have clinically significant consequences, such as increased CVD risk [16]. 

The impact of acute stress on vascular function is often measured in the fasted state, yet during periods of stress, individuals are more likely to overeat and consume unhealthy foods, i.e., fat [17]. Interestingly, fat consumption has been shown to impair endothelial function [18], and endothelial dysfunction has been associated with poorer peripheral vascular responses during stress [19]. 

However, few studies have investigated how fat consumption can influence cerebrovascular function. Initial rodent-based research presented cognitive impairments following a chronic high-fat diet [20]. Furthermore, a two-year population-based study showed that increased total fat consumption associated with a greater incidence of dementia in older adults [21]. As far as we are aware, only two studies to date have investigated the acute ingestion of fat on cerebrovascular outcomes in humans. One reported no change in cerebral perfusion or conductance [22], whereas the other presented decreased CBF to the hypothalamus following fat consumption [23]. Although the mechanisms underlying fat-induced changes in cerebrovascular function are unknown, post-fat impairments in endothelial function [24] as well as increases in inflammation and oxidative stress have been well-documented [25]. These markers are also suggested to play a role in stress-induced changes in vascular function [26].

Given the high prevalence of consumption of high-fat foods during stressful periods [27] and the clinical significance of a healthy cerebrovascular response during stress, in the present study we investigated the impact of fat consumption on cerebrovascular responses during mental stress in healthy young adults. More specifically, we assessed changes in PFC cerebral haemodynamics during a laboratory-based mental stress task following a high- and low-fat meal. We also assessed upstream macrovasculature, by measuring common carotid artery (CCA) vasodilation and blood flow up to 90 min following stress, which is in line with previously evidenced stress-induced and fat-induced impairments in endothelial function. We further quantified changes in mood following high- and low-fat meals and following stress. We hypothesised that a high-fat meal would impair cerebrovascular responses to stress. 

## 2. Materials and Methods

### 2.1. Participants

Healthy, young (age range inclusion: 18–45 years) participants (*n* = 21, 11 male, 10 female), were recruited through email and poster advertisements, and all gave written informed consent prior to participation in the study. Females were tested during the early-follicular phase of the menstrual cycle to control for the effect of menstrual hormones. Inclusion criteria were non-smokers, no history of disease, no allergies or intolerances, and no use of dietary supplements or long-term medication. This study was approved by the University of Birmingham Ethics Committee (ERN17_1755D). 

### 2.2. Procedure

The present study was a cross-over intervention study, with two laboratory visits at least a week apart for males, and approximately a month apart for females. The order of dietary conditions was randomised and counterbalanced. Participants visited the lab at 08:00 h and were asked to refrain from food 12 h before, and from alcohol, vigorous exercise, and caffeine 24 h before each testing session. We also requested that participants followed a similar diet for 24 h prior to each visit. Pre-intervention peripheral vascular measurements were assessed (data reported elsewhere: [28]) as well as common carotid artery (CCA) diameter and blood flow, prior to consumption of a high-fat or low-fat meal. After 1 h 15 min, prefrontal cortical haemodynamics were assessed using near-infrared spectroscopy (NIRS) during an 8-min rest (Rest) and during an 8-min mental stress task (Stress) (Figure 1). Cardiovascular activity was also recorded throughout rest and stress. Immediately following stress, CCA diameter and blood flow were assessed (Post-10). CCA diameter and blood flow were also measured 30 min (Post-30) and 90 min (Post-90) following stress. Each session lasted 5 h, and participants were debriefed following completion of both visits.

### 2.3. Meal Interventions

Both meals were prepared just before consumption, and fresh ingredients were bought within 24 h of each session. The calorie-matched meals consisted of a high-fat meal (HFM, 56.5 g fat) and a low-fat meal (LFM, 11.4 g fat). Nutrients were closely matched, apart from carbohydrate quantity (Table 1). Each meal was consumed within 20 min (excluding 7 participants who consumed approx. 90% of the meals), and no adverse side effects were reported. 

### 2.4. Mental Stress Task

The 8-min paced-auditory-serial-addition-task (PASAT) was used to induce mental stress, shown to have good test–retest reliability and to perturb the cardiovascular system [29]. The task requires participants to add two sequentially presented, single-digit numbers, adding the number presented to the previous number they heard. The time interval between numbers was reduced throughout the task. Elements of social evaluation, punishment, time pressure, and competition (detailed in [28]) were included, shown to enhance the provocativeness of the task [30]. Immediately following the task, participants were asked to verbally rate how difficult, stressful, competitive, and enjoyable they found the task, and to what extent they were trying to perform well, scored on a 7-point scale ranging from 0 ‘not at all’ to 6 ‘extremely’. Following both visits, participants were informed about the deception of the task. 

### 2.5. Cardiovascular Activity

Systolic (SBP) and diastolic (DBP) blood pressure were measured using a Finometer (Finapres Medical Systems, Amsterdam, The Netherlands). A small cuff was placed around the intermediate phalanx of the middle finger, and continuous data were recorded via a Power1401 (CED) connected to a computer programmed in Spike2. 

An ambulatory monitor (VU-AMS) and 7 Ag/AgCl spot electrodes (Invisatrace, ConMed Corporation; Largo, FL, USA) recorded electrocardiographic and impedance cardiographic signals continuously, in accordance with published guidelines [31,32]. Sixty-second ensemble averages were used to determine heart rate (HR, bpm), heart rate variability (HRV, ms), pre-ejection period (PEP, ms), and stroke volume (SV, mL) as measures of sympathetic and parasympathetic activity. Cardiac output (CO, L/min) was calculated as (HR × SV)/1000.

### 2.6. Prefrontal Cortical Haemodynamics

Near-infrared spectroscopy (NIRS, NIRO-200NX, Hamamatsu Photonics KK, Shizuoka, Japan) was used to assess prefrontal cortical haemodynamics. The NIRS device measures changes in chromophore concentrations of oxyhaemoglobin (O_2_Hb) and deoxyhaemoglobin (HHb), providing depth-resolved measures of tissue oxygen saturation (total oxygenation index, TOI) and tissue haemoglobin content (relative value of total haemoglobin normalised to the initial value, nTHI). Probes were positioned over the left pre-frontal site and secured to the head with a black headband. Probes were enclosed in light-shielding rubber housing that maintained emitter-to-detector optode spacing (4 cm), and signals were acquired at a sample interval of 0.2 s (5 Hz). NIRS was assessed during 8 min of rest and 8 min of stress. Measures of TOI, nTHI, O_2_Hb, and HHb were averaged to provide 1 value for each minute of rest and stress. 

### 2.7. Common Carotid Artery Diameter and Blood Flow

Duplex ultrasound was used to assess common carotid artery (CCA) diameter and blood flow. A 15–4 Mhz (15L4 Smart MarK^TM^; Terason, Burlington, MA, USA) transducer was attached to a Terason Duplex Ultrasound System (Usmart 3300 NexGen Ultrasound; Terason). This was combined with wall-tracking and automatic edge-detection software (Cardiovascular Suite, Quipu; Via Moruzzi, Pisa, Italy), which allows for continuous measurement of diameter and blood velocity. Following 10 min of supine rest, the participant was asked to turn their head and neck slightly to the left side. Then, a 2-min recording of the right CCA was obtained. All file images were analysed by a trained researcher, blinded to condition and measurement details. Analysis allows estimation of resting arterial diameter and calculation of arterial blood flow based on a time-average across 2 min of the recording. 

### 2.8. Mood Questionnaire

Mood was assessed with a short form of the Profile of Mood States (POMS) questionnaire [33], calculating 6 constructs: tension–anxiety, anger–hostility, vigour–activity, fatigue–inertia, confusion–bewilderment, and depression–dejection. Participants were asked to rate on a 5-point scale (1 = not at all, 5 = extremely), how they felt at that precise moment. Total mood disturbance (TMD) was calculated by summing all negative items (tension, anger, fatigue, confusion, and depression) and subtracting the positive (vigour) score [34]. POMS questionnaires were completed at pre-intervention baseline (Baseline), post-intervention rest (Rest), immediately following stress (Stress), and 30 and 90 min post-stress (Post-30 and Post-90). 

### 2.9. Data Reduction and Statistical Analysis

NIRS and cardiovascular measures were averaged per minute of assessment for the Rest and Stress periods. For the NIRS variables, the eight rest values were then averaged to one resting baseline value, and reactivity scores during stress were calculated as Stress minus Rest, for minutes 2, 4, 6, and 8 of stress (corresponding to Stress 1, Stress 2, Stress 3, and Stress 4, respectively). 

All data were statistically analysed using IBM SPSS software (version 25). Task perceptions and PASAT scores were compared between visits using a one-way repeated measures ANOVA. Cardiovascular variables were analysed using a two-way repeated measures ANOVA with condition (HFM, LFM) and time (Rest, Stress 1, Stress 2, Stress 3, Stress 4) as within-subject factors. NIRS variables at rest and during stress (8 min averaged) were compared using separate one-sample t-tests for both conditions. This was the most appropriate statistical approach given that the resting values were 0, so there is no variability around the mean. We then further analysed the NIRS variables using a two-way repeated measures ANOVA with condition (HFM, LFM) and time (Stress 1, Stress 2, Stress 3, Stress 4) as within-subject factors. CCA diameter and blood flow were analysed using a 2-condition (HFM, LFM) by 5-time (Baseline, Rest, Post-10, Post-30, Post-90) repeated measures ANOVA. TMD was similarly analysed using a 2-condition (HFM, LFM) by 5-time (Baseline, Rest, Stress, Post-30, Post-90) repeated measures ANOVA. Where appropriate, pairwise comparisons using Bonferroni correction were conducted as post-hoc analyses. All values reported in text, tables, and graphs are mean ± standard deviation. Occasional missing data are reflected in the reported ‘n’ values, and include n − 1 due to VU-AMS malfunction, n − 2 due to finapress malfunction, and n − 2 due to NIRS malfunction. All statistical tests were also carried out excluding 7 participants who did not complete both meals; however, as the results were similar to the analyses with the full sample, all participants were included to maximise power. For all analyses, significance was set at *p* < 0.05. 

## 3. Results

### 3.1. Participant Characteristics

Participants (*n* = 21) ranged from 20 to 30 years old (22.1 ± 2.7 years old), had a healthy BMI (23.6 ± 3.1 kg/m^2^), and identified as either white European ethnicity (*n* = 19) or Asian ethnicity (*n* = 2). Participants self-reported to be physically active and have a healthy habitual diet (daily energy: 1576.5 ± 418.9 Kcal, fat: 59.4 ± 18.9 g, saturated fat: 21.3 ± 6.5 g, carbohydrate: 185.5 ± 57.5 g, sugars: 87.3 ± 42.5 g, fibre: 14.1 ± 5.7 g, protein: 74.3 ± 25.1 g, fruit & vegetables: 5.7 ± 3.3 portions [28]). Resting cardiovascular activity is displayed in Table 2. There were no significant differences in BP, HRV, PEP, and CO between conditions at rest (all *p* > 0.261), although there was a significant difference in post-intervention/pre-stress resting HR between conditions (*p* = 0.027). However, there was no significant difference HR between conditions at the previous pre-intervention timepoint (data shown in [28]). 

### 3.2. Mental Stress Task Ratings

Two-condition (HFM, LFM) ANOVAs revealed no significant difference in PASAT score between conditions (*n* = 21, *p* = 0.544), and participants perceived the task as equally difficult, stressful, competitive, and enjoyable, and tried to perform well to the same extent (all *p* > 0.576) after both high-fat and low-fat meals (Table 3).

### 3.3. Cardiovascular Responses during Mental Stress

Separate 2-condition (HFM, LFM) × 5-time (Rest, Stress 1, Stress 2, Stress 3, Stress 4) ANOVAs revealed an overall time effect for HR (*n* = 20, *p* < 0.001), HRV (*n* = 20, *p* < 0.001), PEP (*n* = 20, *p* < 0.001), CO (*n* = 20, *p* < 0.001), SBP (*n* = 19, *p* < 0.001), and DBP (*n* = 19, *p* < 0.001) (Figure 2). Post-hoc analyses are displayed on Figure 2 (data reported as the change during mental stress relative to rest). There were no significant condition or condition × time interaction effects for HR, HRV, PEP, CO, SBP, and DBP (all *p* > 0.207). 

### 3.4. Prefrontal Cortical Haemodynamics during Mental Stress

One sample *t*-tests revealed that TOI was significantly greater during stress compared to rest in the LFM condition (*p* = 0.005) but not the HFM condition (Figure 3). There were no significant differences in nTHI during stress compared to rest in both conditions.

Separate 2-condition (HFM, LFM) × 4-time (Stress 1, Stress 2, Stress 3, Stress 4) ANOVAs revealed an overall condition effect (*n* = 19, *p* = 0.019) for TOI (Figure 3). Post-hoc analyses revealed that TOI was higher in the LFM condition compared to the HFM condition. However, there were no significant time or condition × time interaction effects for TOI (both *p* > 0.099). There were no significant time, condition, or condition × time interaction effects for nTHI (all *p* > 0.061). 

One sample *t*-tests revealed that O_2_Hb and HHb were significantly different during stress compared to rest in both conditions (both *p* < 0.001) (Figure 4). 

A 2 × 4 ANOVA revealed an overall condition effect (*n* = 19, *p* = 0.048) for O_2_Hb (Figure 4). Post-hoc analyses revealed that O_2_Hb was higher in the LFM condition compared to the HFM condition. There were no significant time nor time × condition interaction effects for O_2_Hb (all *p* > 0.088). A time 2 × 4 ANOVA revealed an overall time effect (*n* = 19, *p* = 0.002) for HHb. Post-hoc analyses revealed that HHb was lower during Stress 2 and Stress 3 compared to Stress 1. There were no condition nor time × condition interaction effects for HHb (all *p* > 0.217). 

### 3.5. Common Carotid Arterial Diameter and Blood Flow Following Mental Stress

CCA diameter and blood flow are reported in Table 4. There were no significant differences in CCA diameter (*p* = 0.561), anterograde blood flow (*p* = 0.698), and retrograde blood flow (*p* = 0.370) between conditions at pre-intervention baseline. A 2-condition (HFM, LFM) × 5-time (Baseline, Rest, Post-10, Post-30, Post-90) ANOVA revealed a significant time effect for CCA diameter (*p* < 0.001). Post-hoc analyses showed that CCA diameter was significantly lower at baseline compared to post-meal rest (*p* = 0.047), 10 min (*p* = 0.047), 30 min (*p* = 0.002), and 90 min post-stress (*p* < 0.001), and CCA diameter at 90 min post-stress was significantly higher than post-meal rest (*p* = 0.026). Furthermore, there was a significant condition × time interaction effect for CCA diameter (*p* = 0.033). Further exploration of this interaction effect revealed that CCA diameter was significantly higher 90 min post-stress in the high-fat condition compared to the low-fat condition (*p* = 0.026). There was no significant condition (*p* = 0.333) effect for CCA diameter. Separate 2-condition × 5-time ANOVAs also revealed no significant time (*p* = 0.535), condition (*p* = 0.357), or condition × time interaction (*p* = 0.924) effect for anterograde blood flow, nor time (*p* = 0.096), condition (*p* = 0.809), or condition × time interaction (*p* = 0.457) effect for retrograde blood flow (Table 4).

### 3.6. Mood Following High and Low-Fat Meal Consumption and Mental Stress

Total mood disturbance (TMD) is presented in Figure 5. There was a significant condition (*p* = 0.013), time (*p* = 0.004), and condition × time interaction effect (*p* = 0.011) for TMD. The time effect revealed that TMD was significantly lower at 90 min post-stress compared to rest (*p* = 0.014) and stress (*p* = 0.030). Furthermore, the condition effect revealed that TMD was overall greater in the high-fat condition compared to the low-fat condition (*p* = 0.013). Finally, as shown in Figure 5, the condition × time interaction effect revealed a significantly higher TMD in the high-fat condition compared to the low-fat condition at post-intervention rest (*p* = 0.003) and immediately following stress (*p* = 0.041). 

## 4. Discussion

The current study showed that mental stress induced increases in HR, CO, SBP, and DBP, decreases in HRV and PEP, and increases in PFC tissue oxygenation (as indexed via changes in TOI and O_2_Hb volume). Following fat consumption (HFM condition), stress-induced increases in PFC tissue oxygenation were attenuated, yet there were no differences in the cardiovascular responses to stress. These cardio/cerebrovascular changes were observed despite no significant differences in stress task perceptions or performance between conditions, indicating a consistent stress experience between visits. We further observed no effect of fat consumption or stress on resting CCAblood flow, whilst CCA diameter increased following consumption of both meals. Consumption of fat influenced mood (TMD) at rest and immediately post-stress, suggesting that fat consumption may negatively affect mood. Taken together, these findings indicate that fat consumption alters cerebral haemodynamic activity while completing a mentally stressful task, potentially via impaired cerebral perfusion to the PFC as a result of fat-induced alterations in CBF regulation. 

Our observation that mental stress increases PFC tissue perfusion (by virtue of increased TOI and O_2_Hb, and decreased HHb) is in line with previous findings that have reported elevated CBF during such stress [10,12]. The increase in CBF is likely to be mediated in part by the systemic increase in CO shown during mental stress, driven by stress-induced elevations in HR (Figure 2). CO is a key independent factor influencing CBF [35], with changes in CO shown to be correlated with CBF at rest and during exercise [36]. In addition, the observed stress-induced increases in BP (SBP and DBP increased by ~20 mmHg) would also contribute to elevated CBF, as even with BP-induced adjustments to cerebrovascular resistance via cerebral autoregulation, CBF will be affected by the large magnitude of observed BP changes [37]. Moreover, our findings are consistent with those of Brindle et al. (2018) [38], where the same stress task resulted in similar changes in BP and increased the TCD-based measures of CBF (i.e., increased middle cerebral blood velocity). Another mechanism by which cerebral perfusion could increase during this stress task is via neurovascular coupling, due to the increased neural activation related to the cognitive demand of the mental arithmetic task. Indeed, Shoemaker and colleagues (2019) [10] showed increased CBF (TCD-based measures of middle cerebral blood velocity) during the cognitive tasks they used, and this occurred independently of other key regulators of CBF (i.e., BP and arterial carbon dioxide content). Similarly, a positive correlation has been evidenced between stress perception and CBF (MRI—arterial spin labelling, ASL) [39], which may contribute to the increase in perfusion via neurovascular coupling, given that both physiological and self-reported data showed the task to be very stressful and difficult, and participants reported to be fully engaged with the task in this study. 

Little is known about how fatty acids affect cerebral oxygenation. To our knowledge, this is the first study to show that fat consumption attenuated the increase in PFC tissue oxygenation during stress, indicating that CBF was relatively lower and therefore more oxygen was extracted from the haemoglobin to meet the metabolic demand of the tissue during the task (assuming brain metabolism was similar for the diet conditions). A previous study has also presented a decreased CBF to the hypothalamus following fat consumption at rest [23]. Given the association between reduced cerebral oxygenation and impaired cognitive performance [40], the lower tissue oxygenation we observed here (2 ± 4%; Figure 3A) may have significant implications for brain function. In the present study, there were no differences in CCA resting blood flow (velocity and diameter) approximately 1 h following fat consumption, but it is possible that more subtle regional changes downstream in the cerebrovasculature could have occurred which we did not assess. Furthermore, another study, with comparable fat quantity to the present study, found no change in CBF during rebreathing-induced hypercapnia following fat consumption [22], potentially suggesting some specificity of the fat effect in the context of mental stress. However, differences in methodology for CBF assessments, brain area investigated (i.e., TCD-based cerebral blood velocity [22] vs. fMRI of hypothalamic and insular cortex using ASL [23]), and differences in fat source might also contribute to some of the differences reported. The mechanisms underlying the fat-induced attenuation of cortical blood perfusion during stress are not known. One possibility is that fat consumption affects cerebral metabolism (exchange of primary molecules of oxygen, glucose, and lactate across arterial and venous circulations in the brain) during stress [41]. As neural activity increases, e.g., at the onset of mental stress [42], dendrites rapidly consume oxygen, reducing P_O2_ and oxyhaemoglobin concentration [43]. The resulting shifts in brain metabolism enhance glycolysis in astrocytes and induce a release of lactate, which subsequently causes vasodilation to increase oxygen delivery [41,43]. Therefore, whilst speculative, fat consumption may reduce the metabolic efficiency of the brain by attenuating this shift in metabolism and hence, reducing perfusion during stress. Whilst evidence is limited to support this idea, evidence that obesity and a high-fat diet can alter metabolic-related cerebral signalling and induce neuroinflammation, thus disrupting cognitive function, has been reviewed [44]. Further research is needed to explore this mechanism, for example using broadband NIRS measurements of cytochrome-c-oxidase (CCO) to investigate brain metabolism following fat consumption. Another possible mechanism is that fat intake, and specifically hyperlipidaemia, influences cerebral autoregulation [45]. However, if fat consumption did impair cerebral autoregulation, it would be expected that CBF would increase more to the same stress-induced increase in BP; however, we observed the opposite effect with respect to changes in PFC tissue oxygenation. nTHI responses seem to be attenuated during mental stress following the high-fat meal, albeit non-significantly. Given that nTHI is an index of tissue blood flow (via measures of total haemoglobin volume), these data suggest that CBF responses to stress are reduced under high-fat conditions and that cerebral autoregulation impairment does not play a significant role in the observed responses. Whilst the mechanisms by which fatty acids affect cerebral oxygenation are unclear, the literature on muscle physiology clearly shows that acute fatty acid intake can blunt leg blood flow responses to NG-monomethyl-L-arginine (L-NMMA), an NO synthase inhibitor [46], providing evidence that fatty acid elevation impairs NO-mediated vasodilation in the leg microvasculature. Furthermore, animal models show that acute fat intake induces insulin resistance and subsequent impairments in capillary recruitment and muscle glucose uptake [47,48]. The extent to which some of these mechanisms translate into the brain microvasculature is unclear and needs to be investigated further. 

Interestingly, although fat consumption alters cerebral haemodynamics during mental stress, from 10 to 90 min following stress, no differences in resting carotid arterial blood flow between diets were detectable. It should be noted that stress-induced and fat-induced declines in peripheral vascular function have been well established during the period of 30–90 min post stress [26,49], and was the reason we targeted this timeframe for our post-stress assessments. Given that elevated BP and CO are shown to influence CBF [36], perhaps once these have returned to baseline (~10 min following stress), there is no longer a detectable effect on CBF. Furthermore, we are assessing the upstream macrovasculature (common carotid artery), which supplies the whole brain (as well as some extracranial tissue via the external carotid artery that originates from the CCA), and not specifically the PFC, so more subtle and specific changes might have been missed. 

Finally, we observed that CCA diameter significantly increased following consumption of both meals and was significantly greater after the high-fat meal compared to the low-fat meal only at 90 min post stress. This is possibly driven by cholecystokinin (CCK), a peptide hormone that increases postprandially to stimulate digestion, and has been shown to induce cerebral vasodilation [50]. More specifically, the release of CCK in response to a meal has been shown to trigger local postprandial hyperaemia in the gut and evoke vasodilation in the cerebral vasculature [51]. CCK has also been shown to stimulate neuronal NO synthase and NO release, via intracellular calcium [52], which may also induce vasodilation. Furthermore, there is some evidence that CCK levels are higher following a high-fat meal compared to a high-carbohydrate meal [53], which may explain the increase in CCA diameter in the high-fat meal condition in the present study. However, as there was no change in CCA blood flow, future research should continue this investigation, utilising assessments of the internal carotid artery, to assess the impact of fat consumption on resting CBF.

Fat consumption had a significant impact on mood in the present study, shown by a greater mood disturbance at rest and immediately following stress in the high-fat condition compared to the low-fat condition. When exploring the individual constructs that are used to calculate total mood disturbance, it was particularly fatigue which was significantly higher following the high-fat meal compared to the low-fat meal, which is in line with previous research [54]. Whilst the relationship between fat consumption and mood outcomes is currently unclear [55], previous evidence suggests that high-fat feeding leads to negative emotional states and even increased stress sensitivity in rodent models [56]. Furthermore, it is widely recognised that the PFC plays a central role in emotion regulation via efferent projections to limbic areas (responsible for emotional responses) [57]. As such, there might be a link between the observed decline in PFC oxygenation during stress and the decline in mood that follows, although this needs to be further addressed in future studies. Therefore, whilst individuals may seek comfort through consumption of high-fat foods when stressed, such food choices may further worsen mood, increase fatigue, and affect an individual’s ability to cope with stress, possibly via disturbances in PFC oxygenation. 

### Limitations

One of the potential limitations of the current study was that the meals were not tailored to individual metabolic rate. Yet, previous studies have shown that a similar fat content (50 g) is sufficient to impact vascular function, and the current study was in line with a similar study showing fat consumption impairs endothelial function [58]. Secondly, our study has a moderate sample size; nevertheless, a crossover design was employed, and effect sizes for non-significant findings were found to be small, suggesting that a lack of power is not likely to drive these results. Furthermore, as these analyses are secondary, no a priori power calculations were undertaken. Yet, based on the effect size of the condition effect revealed in TOI (0.27), with a sample of 21 participants and alpha set at 0.05, we were able to detect a power of 82%. Importantly, this is the first study to investigate the impact of fat consumption on cerebrovascular responses in a sample that includes females, which is more ecologically valid. Finally, it would have been ideal to assess changes in blood flow in the upstream carotid artery during stress to have a more complete picture of cerebral regulation, but this would be unreliable due to significant movement and positioning of the participant. We also noted that it could significantly interfere with the completion of the stress task itself. Future studies should use combined approaches to assess macro- and microvasculature significantly, as well as explore regional differences across the brain [59], for example, by using techniques such as transcranial doppler and arterial spin labelling (ASL) magnetic resonance imaging (MRI), in addition to NIRS and ultrasound. Importantly, using specific technical approaches such as broadband NIRS and functional MRI would allow for a simultaneous assessment of vascular and neuronal metabolic responses during stress, which may shed light on the mechanisms by which fat reduces cortical blood perfusion during stress. 

## 5. Conclusions

This is the first study to explore the relationship between fat consumption and cerebral dynamics during mental stress, providing, for the first time, evidence that fat consumption impairs PFC perfusion during stress. Experiencing stress is tightly associated with consuming high-fat foods [17]. This, combined with the high prevalence of stress and obesity in our societies, and further associations with cognitive decline later in life, makes it an important area of research to inform our dietary choices during periods of enhanced stress. 

## Figures and Tables

**Figure 1 nutrients-15-03969-f001:**
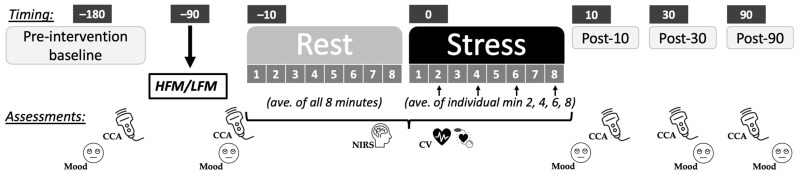
Protocol diagram.

**Figure 2 nutrients-15-03969-f002:**
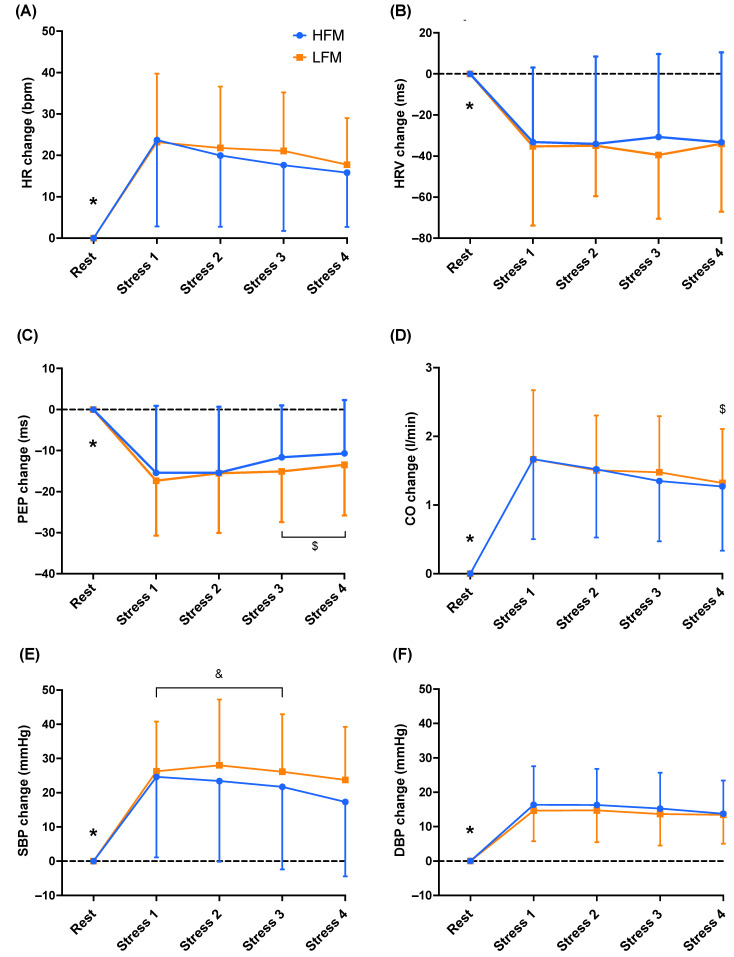
Time course of cardiovascular responses (HR (**A**), HRV (**B**), PEP (**C**), CO (**D**), SBP (**E**), DBP (**F**)), during rest and stress following either an HFM or LFM. Data are presented as reactivity mean ± standard deviation. *n =* 20 (**A**–**D**)/19 (**E**,**F**). * Significantly different compared to Stress 1, 2, 3, and 4, $ significantly different compared to Stress 1, & significantly different compared to Stress 4. HR: heart rate, HRV: heart rate variability, PEP: pre-ejection period, CO: cardiac output, SBP: systolic blood pressure, DBP: diastolic blood pressure, HFM: high-fat meal, LFM: low-fat meal.

**Figure 3 nutrients-15-03969-f003:**
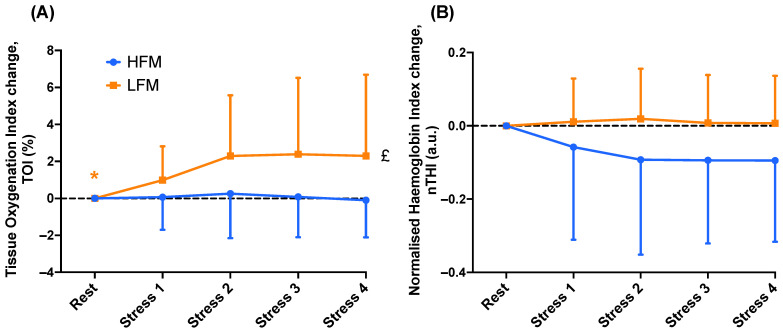
Time course of prefrontal cortical haemodynamics (TOI (**A**) & nTHI (**B**)) during rest and stress following either an HFM or LFM. Data are presented as reactivity mean ± standard deviation. *n =* 19. * Significantly different compared to stress in the LFM condition (*t*-test), £ significantly higher following LFM compared to HFM. TOI: tissue oxygenation index, nTHI: normalised haemoglobin index, HFM: high-fat meal, LFM: low-fat meal.

**Figure 4 nutrients-15-03969-f004:**
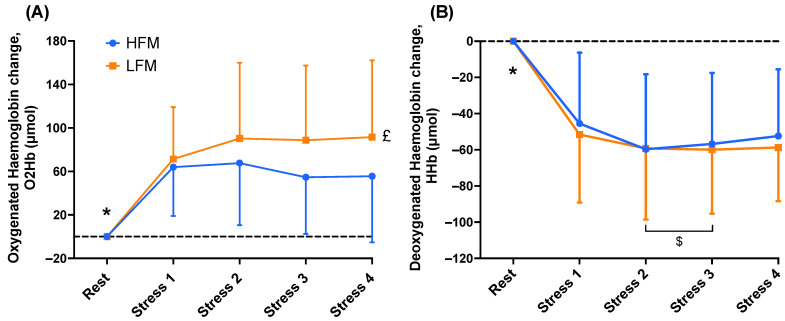
Time course of prefrontal cortical haemodynamics (O_2_Hb (**A**) & HHb (**B**)) during rest and stress following either an HFM or LFM. Data are presented as reactivity mean ± standard deviation. *n =* 19. * Significantly different compared to stress (*t*-test), £ significantly higher following LFM compared to HFM, $ significantly different compared to Stress 1. O_2_Hb: oxygenated haemoglobin change, HHb: deoxygenated haemoglobin change, HFM: high-fat meal, LFM: low-fat meal.

**Figure 5 nutrients-15-03969-f005:**
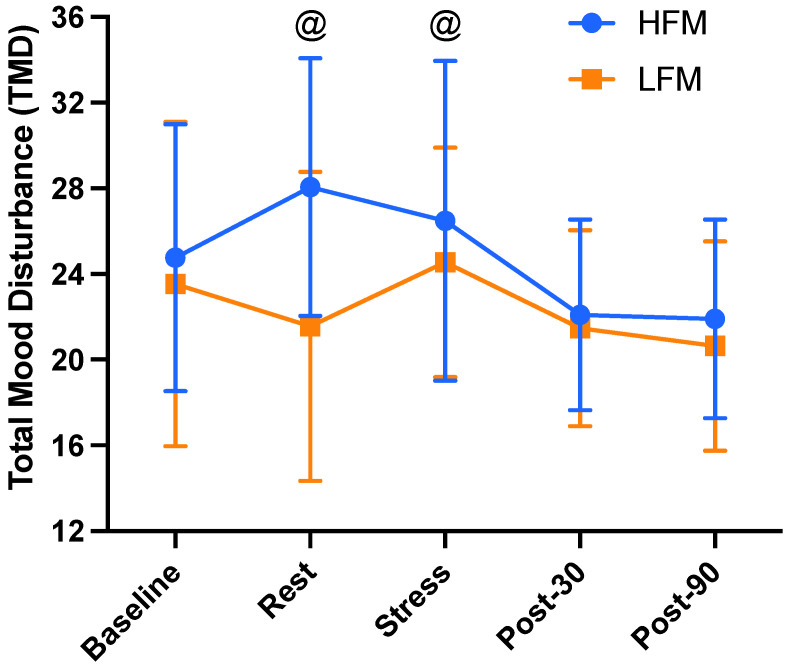
Time course of total mood disturbance at baseline, rest, immediately post-stress, and 30 and 90 min post-stress, following either an HFM or LFM. Data are presented as mean ± standard deviation. TMD = (Tension + Anger + Fatigue + Depression + Confusion) − Vigour. *n =* 18. @ Significant difference between HFM and LFM at these time points. TMD: total mood disturbance, HFM: high-fat meal, LFM: low-fat meal.

**Table 1 nutrients-15-03969-t001:** Nutrient composition of the high-fat and low-fat meals.

Meal Type	High-Fat Meal ^1^	Low-Fat Meal ^2^
Energy (Kcal)	891.00	886.00
Fat (g)	56.50	11.40
Saturated fat (g)	35.10	5.55
Carbohydrate (g)	65.00	160.10
Sugars (g)	20.2	19.40
Fibre (g)	2.40	5.90
Protein (g)	29.85	33.30
Salt (g)	2.00	2.53

^1^ This meal consists of 2 butter croissants with 10 g salted butter, 1.5 slices of cheese, and 250 mL whole milk. ^2^ This meal consists of 4 slices of white bread with 30 g Philadelphia light spread, 90 g SO organic cornflakes, and 250 mL semi-skimmed milk.

**Table 2 nutrients-15-03969-t002:** Mean ± SD resting participant characteristics in the high-fat and low-fat conditions.

	High-Fat Meal	Low-Fat Meal
SBP (mmHg)	127 ± 18	123 ± 14
DBP (mmHg)	52 ± 11	50 ± 8
HR (bpm)	67 ± 9	64 ± 8 *
HRV (ms)	75 ± 50	77 ± 41
PEP (ms)	99 ± 23	99 ± 18
CO (L/min)	7 ± 2	6 ± 2

*n* = 19 (BP)/20 (HR, HRV, PEP, CO). * *p* < 0.05. SBP: systolic blood pressure, DBP: diastolic blood pressure, HR: heart rate, HRV: heart rate variability, PEP: pre-ejection period, CO: cardiac output.

**Table 3 nutrients-15-03969-t003:** Mean ± SD task performance and ratings in each meal condition.

Task Ratings	High-Fat Meal	Low-Fat Meal
PASAT Score	141 ± 34	138 ± 35
Perceived difficulty	4.8 ± 0.6	4.7 ± 0.7
Perceived stressfulness	4.9 ± 0.9	4.7 ± 0.7
Perceived competitiveness	4.3 ± 1.2	3.9 ± 1.4
Perceived enjoyment	2.0 ± 1.2	1.5 ± 1.1
Perception of trying to perform well	5.0 ± 0.9	5.1 ± 1.0

Note: Task ratings scored from 0–6 and PASAT score/228. *n* = 21.

**Table 4 nutrients-15-03969-t004:** Mean ± SD common carotid arterial diameter and blood flow following mental stress.

	High-Fat Meal					Low-Fat Meal				
Timepoint	Baseline	Rest	Post-10	Post-30	Post-90	Baseline	Rest	Post-10	Post-30	Post-90
Diameter (mm)	6.69 ± 0.58	6.76 ± 0.56 ^&^	6.75 ± 0.59 ^&^	6.79 ± 0.58 *	6.87 ± 0.56 *^£^	6.67 ± 0.56	6.75 ± 0.55 *	6.76 ± 0.51 *	6.76 ± 0.51 *	6.76 ± 0.53 *^£^
Anterograde blood flow (cm^3^/min)	695.34 ± 205.10	717.03 ± 195.22	707.69 ± 193.96	709.37 ± 199.11	699.49 ± 140.94	683.40 ± 142.39	712.70 ± 165.50	680.56 ± 182.32	699.44 ± 186.51	668.28 ± 149.80
Retrograde blood flow (cm^3^/min)	−0.72 ± 0.94	−1.83 ± 1.97	−1.06 ± 1.77	−2.28 ± 3.42	−2.63 ± 3.27	−1.53 ± 4.36	−1.18 ± 1.52	−1.06 ± 1.40	−2.64 ± 5.45	−1.51 ± 2.16

* Significantly different compared to baseline, ^&^ significantly different compared to post-90, ^£^ significantly different between conditions. *n* = 21.

## Data Availability

The data presented in this study are available on request from the corresponding author.

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
