# Peer review of "Fat Consumption Attenuates Cortical Oxygenation during Mental Stress in Young Healthy Adults"

_nutrients, 2023, doi:10.3390/nu15183969_

Round 1

Reviewer 1 Report

By this manuscript, authors willing to evaluate differences in cerebrovascular responses to stress to high and low-fat food intake.

Paper is clear and fluent; topic is nicely presented supported by appropriate literature references.

Data collected are enough to support conclusions.

No any concern from my side

Reviewer 2 Report

This randomized crossover trial offers significant insights into the impact of a high-fat diet and mental stress on cerebral blood flow. There are, however, certain areas within the manuscript that could benefit from further enhancement to improve its overall quality.

1. The manuscript could be enriched by considering, discussing, and controlling for additional variables that may potentially influence cortical oxygenation and carotid blood flow. These variables could include dietary habits, body weight, and physical activity levels. Could the authors elucidate upon whether these factors were balanced between groups?

2. Given the somewhat limited sample size used in this study, it would be beneficial for the authors to provide an explanation for their choice of sample size. In addition, a more detailed discussion surrounding the reliability and potential limitations of the statistical analyses carried out would further strengthen the study.

3. In the discussion section, a more comprehensive explanation and examination of the experimental results would be advantageous. It would be particularly interesting to explore the potential underlying mechanisms by which high-fat consumption might influence cerebral blood supply. A comparison with findings from previous research would also provide valuable context and support for your results.

In light of the points raised above, I recommend that the authors undertake a minor revision.
